# Unravelling key enzymatic steps in C-ring cleavage during angucycline biosynthesis

Somayah S. Elsayed [1,7 ✉], Helga U. van der Heul[1,7], Xiansha Xiao[2], Aleksi Nuutila [3], Laura R. Baars[4], Changsheng Wu[5], Mikko Metsä-Ketelä[3] & Gilles P. van Wezel [1,6 ✉]

Angucyclines are type II polyketide natural products, often characterized by unusual structural rearrangements through B- or C-ring cleavage of their tetracyclic backbone. While the enzymes involved in B-ring cleavage have been extensively studied, little is known of the enzymes leading to C-ring cleavage. Here, we unravel the function of the oxygenases involved in the biosynthesis of lugdunomycin, a highly rearranged C-ring cleaved angucycline derivative. Targeted deletion of the oxygenase genes, in combination with molecular networking and structural elucidation, showed that LugOI is essential for C12 oxidation and maintaining a keto group at C6 that is reduced by LugOII, resulting in a key intermediate towards C-ring cleavage. An epoxide group is then inserted by LugOIII, and stabilized by the novel enzyme LugOV for the subsequent cleavage. Thus, for the first time we describe the oxidative enzymatic steps that form the basis for a wide range of rearranged angucycline natural products.

[1] Department of Molecular Biotechnology, Institute of Biology, Leiden University, Sylviusweg 72, 2333BE Leiden, The Netherlands. [2] Department of Structural Biology, Van Andel Institute, Grand Rapids, MI, USA. [3] Department of Life Technologies, University of Turku, Tykistökatu 6, FIN-20014 Turku, Finland. [4] Department of Systems Pharmacology and Pharmacy, Leiden Academic Centre for Drug Research, Leiden University, Einsteinweg 55, 2333CC Leiden, The Netherlands. [5] State Key Laboratory of Microbial Technology, Institute of Microbial Technology, Shandong University, 266237 Qingdao, P.R. China. [6] Department of Microbial Ecology, Netherlands Institute of Ecology (NIOO-KNAW), Droevendaalsesteeg 10, 6708PB Wageningen, The Netherlands. [7]These authors contributed equally: Somayah S. Elsayed, Helga U. van der Heul. ✉email: s.elsayed@biology.leidenuniv.nl; g.wezel@biology.leidenuniv.nl

Angucyclines are a group of glycoside natural products whose aglycones, which are termed angucyclinones, are derived from a benz[a]anthracene skeleton biosynthesized via the polyketide pathway (Fig. 1a)[1]. Collectively they represent the largest group of type II polyketide synthase (PKS) derived natural products, with more than 400 documented members, of which around 45% have biological activity[2]. Many have antimicrobial and/or anticancer activity, but also other bioactivities including acting as gastric mucosal protectant, vasodilator, glutamate receptor agonist, platelet aggregation inhibitor, or antidiabetic[3]. The attention drawn to angucyclines is not only due to their potent biological activities, but also due to their diverse chemical scaffolds, leading to the discovery of new synthetic and biosynthetic routes and enriching the chemical space for drug discovery.

All angucyclin(on)es identified so far have been isolated from Actinobacteria. Type II PKS enzymes use one acetyl-CoA and nine malonyl-CoA molecules to eventually produce the characteristic angular tetracyclic backbone (Fig. 1a)[3,4]. The first stable intermediate in multiple angucyclines biosynthetic routes is prejadomycin (1)[5], which is formed through enzymatic hydrolysis and decarboxylation of an acyl carrier protein (ACP)-bound intermediate (Fig. 1a)[6]. On other pathways, UWM6 (2), which is unstable and can spontaneously oxidize to rabelomycin (3)[7], is the key precursor towards pathway end-products, like in urdamycin[4] and gaudimycin[8] biosynthesis. UWM6 is formed through spontaneous hydrolysis of the ACP-bound intermediate. Modifications to the angucyclinone backbone such as oxidation, reduction, and glycosylation are catalyzed by post-PKS tailoring enzymes, leading to a great diversity in chemical structures[9]. One

**Fig. 1 Angucyclines produced by *Streptomyces* sp. QL37. a** The early biosynthetic steps leading to the production of the angucyclinone backbone. **b** Structures of some angucyclinones previously identified from *Streptomyces* sp. QL37.

of the most intriguing modifications to angucyclinones is oxidative ring cleavage and rearrangement, resulting in metabolites that lack the typical benz[a]anthracene structure. Cleavage of the B-ring followed by ring contraction and other rearrangements was observed in, for example, gilvocarcins, kinamycins, fluostatins, and lomaiviticins[10–13]. Several studies were consequently dedicated to the total chemical synthesis of these metabolites, and also to the understanding of the underlying enzymology and mechanisms of biosynthesis[14–19]. Eventually, it was discovered that a single oxygenase catalyzes B-ring cleavage through an initial C5 hydroxylation followed by Baeyer–Villiger oxidation (BVO) at the C6 keto group, leading to ring opening and rearrangement[15].

In addition, a variety of angucyclinones featuring C-ring cleavage and rearrangement have been described[20–23]; however, the enzymology responsible for this rearrangement is poorly understood. Decades ago, a BVO reaction was proposed for C-ring cleavage, based on a feeding experiment with $^{13}$C- and $^{18}$O-labeled precursors, which showed the introduction of an oxygen to the polyketide chain between C6a and C7[24]. At the time, the enzymes involved were not known. Recently, multiple alternative mechanisms were reviewed based on a global overview of the chemical structures identified thus far, but no experimental evidence has been provided[25]. Thus, despite their wide distribution and high abundance, it is still unclear what the mechanism is that leads to cleavage of the C-ring in angucyclinones.

We previously discovered lugdunomycin (4), a highly rearranged angucycline-derived natural product featuring C-ring cleavage, together with many other rearranged and non-rearranged angucyclinones (Fig. 1b)[23]. The biosynthetic gene cluster (BGC) that specifies the angucyclines in the producing strain Streptomyces sp. QL37 was identified, and deletion of the genes encoding the minimal PKS enzymes prevented the production of any angucyclines. As the introduction of an oxygen at the C-ring cleavage site was earlier proven[24], we focused our current study on the putative oxygenases encoded by the BGC (designated lug). Here we have employed targeted gene deletion and complementation, metabolite purification and structure elucidation, which we combined with extensive metabolome analysis through molecular networking. Our study is the first to show with experimental evidence that angucyclinone C-ring cleavage proceeds through epoxidation and the sequential activity of the enzyme pair, LugOIII and LugOV, belonging to the antibiotic biosynthesis monooxygenase (ABM) family.

## Results and discussion
### Phylogenetic analysis of the lug oxygenases revealed two putative antibiotic biosynthesis monooxygenases.
Bioinformatic analysis of the different genes in the lug BGC returned five (lugOI–OV) which encode putative oxygenases (Fig. 2, Supplementary Information Table S1). Orthologs of all five genes were found in the angucycline producers Streptomyces sp. W007, Streptomyces sp. CB00072 and Streptomyces sp. Go-475[26–29]. Of those three, C-ring fragmented angucyclines were identified from Streptomyces sp. W007 and CB00072. To get a first indication of the functionality of the various LugO enzymes, they were subjected to phylogenetic analysis (Fig. 3). The phylogenetic tree includes known angucycline oxygenases, anthrone oxygenases like TcmH and ActVA-ORF6[30], and Baeyer–Villiger monooxygenases (BVMOs) involved in the biosynthesis of other polyketides, such as MtmOIV and XanO4[31].

LugOI clusters with flavoprotein monooxygenases, like PgaE and UrdE, which act as C12 as well as C12b hydroxylases[5,32,33]. LugOII clusters with PgaM and UrdM which are bifunctional enzymes with an N-terminal flavoprotein monooxygenase domain and a C-terminal short-chain dehydrogenase/reductase (SDR) domain[34,35]. The oxygenase part acts as a 2,3-dehydratase, while the reductase part catalyzes a C6 ketoreduction[6,34]. LugOIII–OV do not cluster closely with oxygenases of known functions. A search in the Pfam database for protein families[36] revealed that LugOIV contains an SDR domain. On the other hand, LugOIII and LugOV contain an antibiotic biosynthesis monooxygenase domain (ABM), although the sequence similarity to the ABM in case of LugOV is quite low (E-value 0.00038). Overall LugOIII and LugOV share 30% aa sequence identity (Supplementary Information Fig. S1), indicating that both might function as monooxygenases. LugOIII and LugOV are distantly related to B-ring cleavage enzymes and anthrone oxygenases (Fig. 3). They also do not have a recognizable binding site for cofactors or metal ions. While anthrone oxygenases do not require any cofactors or metal ions[30], the B-ring cleavage enzymes are flavin dependent, even though they have no recognizable flavin binding site[15].

### Effect of the deletion of lugO genes on the angucycline metabolome in Streptomyces sp. QL37.
To obtain more insights into the roles of the different putative oxygenases in C-ring cleavage of angucyclinones, lugOI–OV were individually deleted in Streptomyces sp. QL37 using a method published previously[37,38]. Briefly, for the generation of the knock-out construct, around 1.5 kb of the upstream and downstream regions of the genes were amplified by PCR from the genomic DNA from Streptomyces sp. QL37 with the primer pairs listed in Supplementary Information Table S3. The DNA fragments were cloned into the multicopy vector pWHM3-oriT, and the engineered XbaI site was used for insertion of the aac(3)IV apramycin resistance cassette flanked by loxP sites between the flanking regions. The construct was conjugated to Streptomyces sp. QL37 using the methylase deficient strain Escherichia coli ET12567/pUZ8002[39,40]. The presence of the loxP recognition sites allowed the efficient removal of the apramycin resistance cassette after the introduction of plasmid pUWLcre expressing Cre recombinase[41].

The lugO mutants, the previously generated minimal PKS lugA–C mutant[23] and the parental wild-type strain were each cultured on both minimal medium (MM) and R5 (R5) agar plates, which were selected for production of lugdunomycin (4) and compounds 10–12, respectively[23]. Subsequently, the metabolites in the agar and the mycelium were extracted with ethyl acetate (EtOAc) and analyzed using liquid chromatography coupled to mass spectrometry (LC-MS). The LC-MS data obtained were processed using MZmine 2[42], and the processed data were submitted to the Global Natural Products Social Molecular Networking web tool (GNPS) for molecular networking[43]. In the end, two networks were generated representing the ions detected in the extracts from the cultures of Streptomyces sp. QL37 and its lugO mutants grown on either MM or R5 (Supplementary Information Fig. S2). This allowed a clear overview of the qualitative and quantitative changes in the metabolome of Streptomyces sp. QL37 upon the deletion of the different lugO genes.

Based on the known metabolites (Supplementary Information Figs. S3–S14), we identified the molecular families for typical non-rearranged angucyclinones as well as their C-ring cleaved rearranged congeners (Figs. 4 and 5). The relative number and intensities of the typical angucyclinones were higher in R5-grown cultures. Conversely, the rearranged metabolites related to lugdunomycin (4) and elmonin (13) were only detected in MM cultures. While compounds 10–12 were previously only detected

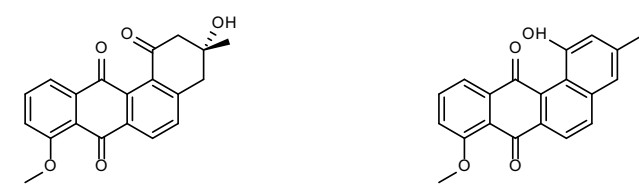

**Fig. 2 Comparison of the *lug* with related BGCs in other *Streptomyces* strains.** The annotated *lugO* genes are highlighted in different colors (*lugOI*, red; *lugOII*, pink; *lugOIII*, yellow; *lugOIV*, green; *lugOV*, cyan).

in R5-grown cultures[23], our in-depth metabolome analysis using molecular networking showed that they could also be produced on MM, but at a much lower level. On both media, all angucycline-related metabolites were detected in cultures of the wild-type strain and the *lugOIV* mutant, although the production levels were generally reduced in the *lugOIV* mutant. This indicates that LugOIV is not directly involved as a post-PKS enzyme, but may have an auxiliary role in promoting angucycline biosynthesis.

**LugOI and LugOII are the early tailoring enzymes towards C-ring cleavage.** Further analysis of the molecular networks showed that deletion of *lugOI* abolished the production of almost all of the non-rearranged angucyclinones and all of the rear-

ranged ones. Among the known angucyclinones from *Streptomyces* sp. QL37, only 8-*O*-methylrabelomycin (**7**) and its derivatives **8** and **14** could still be identified in the *lugOI* mutant (Figs. 4 and 5). In addition, several molecules accumulated, particularly in R5-grown cultures, including **1**, **2**, **3** and angucycline dimers that could be related to **3** (Supplementary Information Fig. S15). The profile of the *lugOI* mutant aligned with the function of several known LugOI homologs, which are early post-PKS enzymes that are required for the C12 hydroxylation of **1** and/or **2**. The non-enzymatically oxidized **3** may further form dimers spontaneously[7,44]. The complementation of the *lugOI* mutant with a biochemically well characterized homolog *pgaE*[34] restored the wild-type metabolomic profile (Supplementary Information Fig. S16), which confirmed that *lugOI* does not have any additional atypical functions.

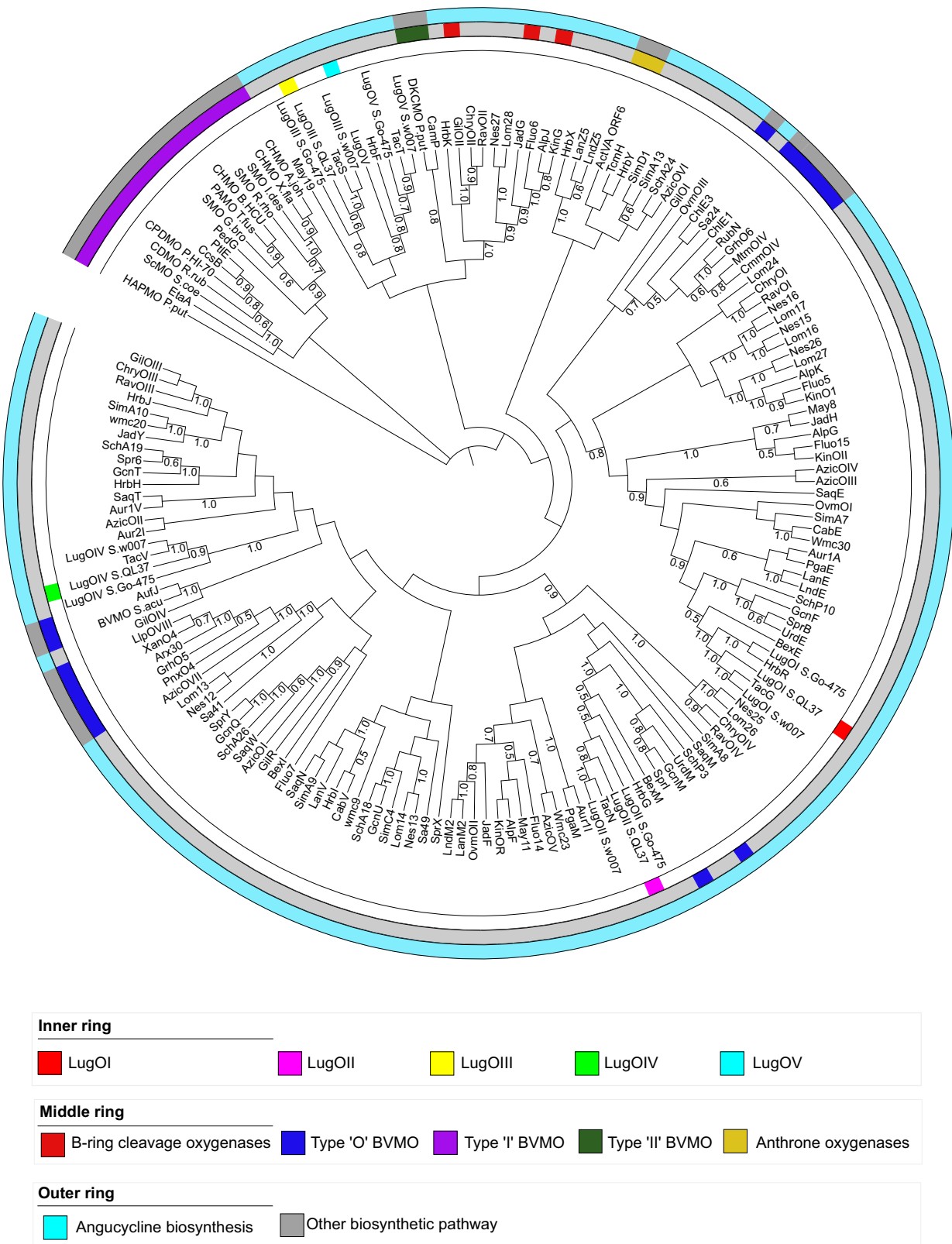

**Fig. 3 Phylogenetic tree of the oxygenases and BVMOs related to angucycline biosynthetic pathways that require aromatic ring opening.** The percentage of replicate trees in which the associated proteins clustered together in the bootstrap test (500 replicates) are indicated next to the branches (in a scale from 0 to 1). Only bootstrap values of >0.5 are shown at the nodes.

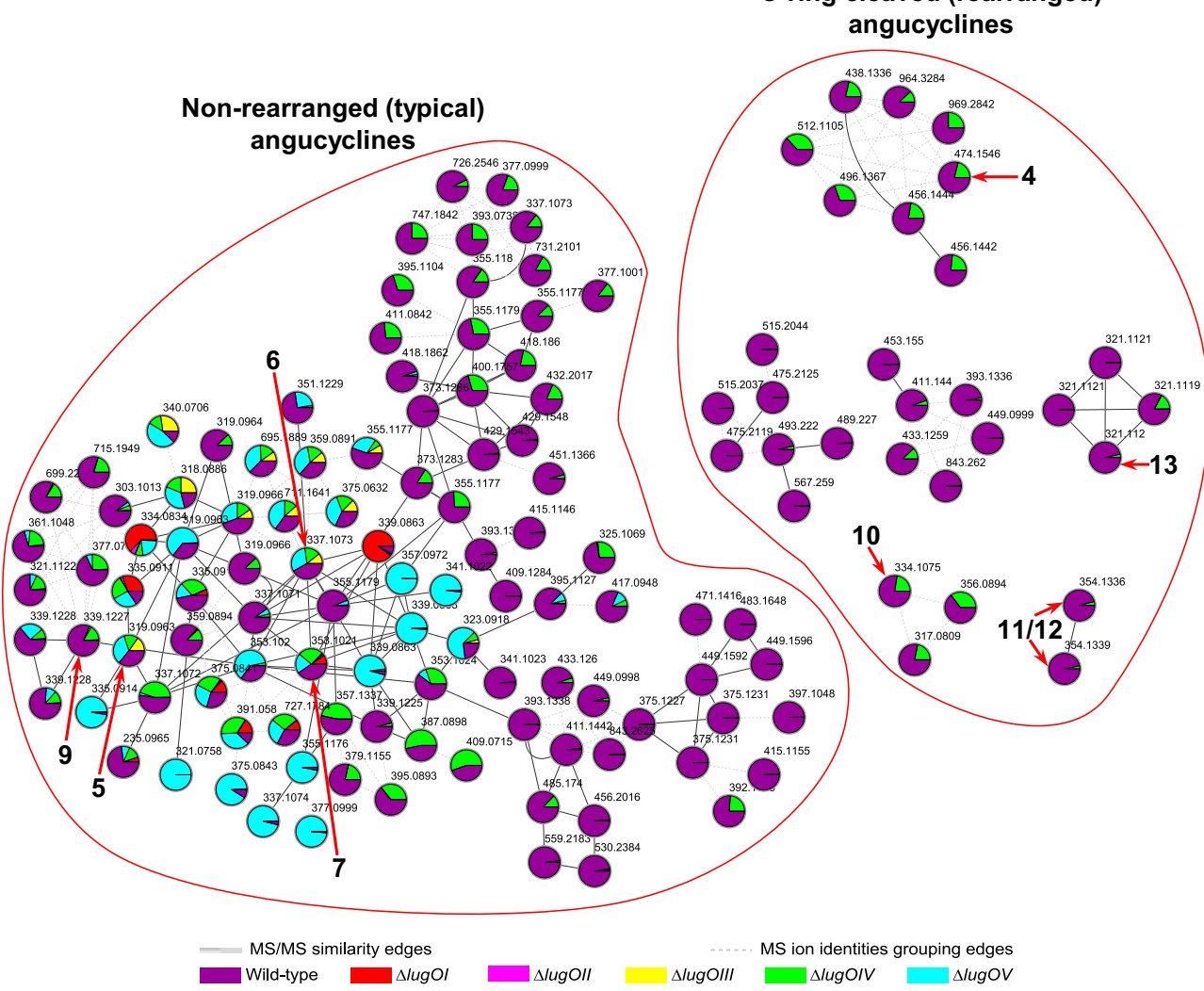

**Fig. 4 Angucycline molecular families identified in the extracts of *Streptomyces* sp. QL37 and its *lugO* mutants, grown on MM agar.** The nodes are labeled by the precursor mass of their ions and pie charts are mapped to the nodes representing the relative intensities of the ions in the different samples. Arrows are used to highlight the metabolites which were identified in the extracts.

Deletion of *lugOII* resulted in a similar metabolomic profile to that of the *lugOI* mutant, but with fewer spontaneously generated angucyclinones, and the absence of **14**. The presence of an intact LugOI in the *lugOII* null mutant may have reduced the number— and sometimes the amount—of shunt metabolites. As mentioned above, LugOII has a high sequence similarity to bifunctional enzymes which have both oxygenase and reductase domains. We propose that the oxygenase domain (LugOIIox) acts as a 2,3-dehydratase resulting in the formation of **1** (Fig. 1), similarly to homologous proteins from related pathways[6,45], which may have affected the production of multiple angucyclinones utilizing **1** as their precursor.

Orthologs of the reductase domain of LugOII (LugOIIred) catalyze a C6 ketoreduction, and the complementation of the *lugOII* mutation with an intact copy restored the production of the C6 ketoreduced metabolites **5**, **6**, and **9**[38]. The absence of the C6 ketoreduced metabolites in the *lugOI* mutant indicates that LugOI is required for the C6 ketoreduction catalyzed by LugOII. Such role is not solely related to the C12 hydroxylation activity of LugOI, since this could be spontaneously induced. We propose that LugOI might prevent the tautomerization of the C6 keto group to the more stable phenol form, which is observed in the

spontaneously generated **3**. LugOIIred is a promiscuous enzyme additionally catalyzing an unprecedented C1 ketoreduction[38], which explains the absence of **14**, and in part **15**, in the *lugOII* mutant. Such activity does not require LugOI, as **14** could still be observed in the *lugOI* mutant.

As almost all of the C-ring cleaved angucyclinones identified so far bear a C6 ketoreduced structure[23,25], a major impact on the metabolome was expected based on the absence of the C6 ketoreduction step upon the deletion of *lugOII*. This was indeed observed in the molecular networks of *Streptomyces* sp. QL37 and its *lugO* mutants (Figs. 4 and 5). C6 ketoreduction is also required for the biosynthesis of some angucycline families like landomycins, urdamycins, and gaudimycins[3]. However, it is not required for the B-ring cleavage of angucyclines, and dehydrorabelomycin was the substrate used in vitro by the B-ring cleavage enzyme GilOII[15].

**LugOIII and LugOV are involved in C-ring cleavage reactions.** To obtain more insights into the involvement of LugOIII and LugOV in C-ring cleavage, we continued investigating the changes in the angucycline metabolome upon the deletion of

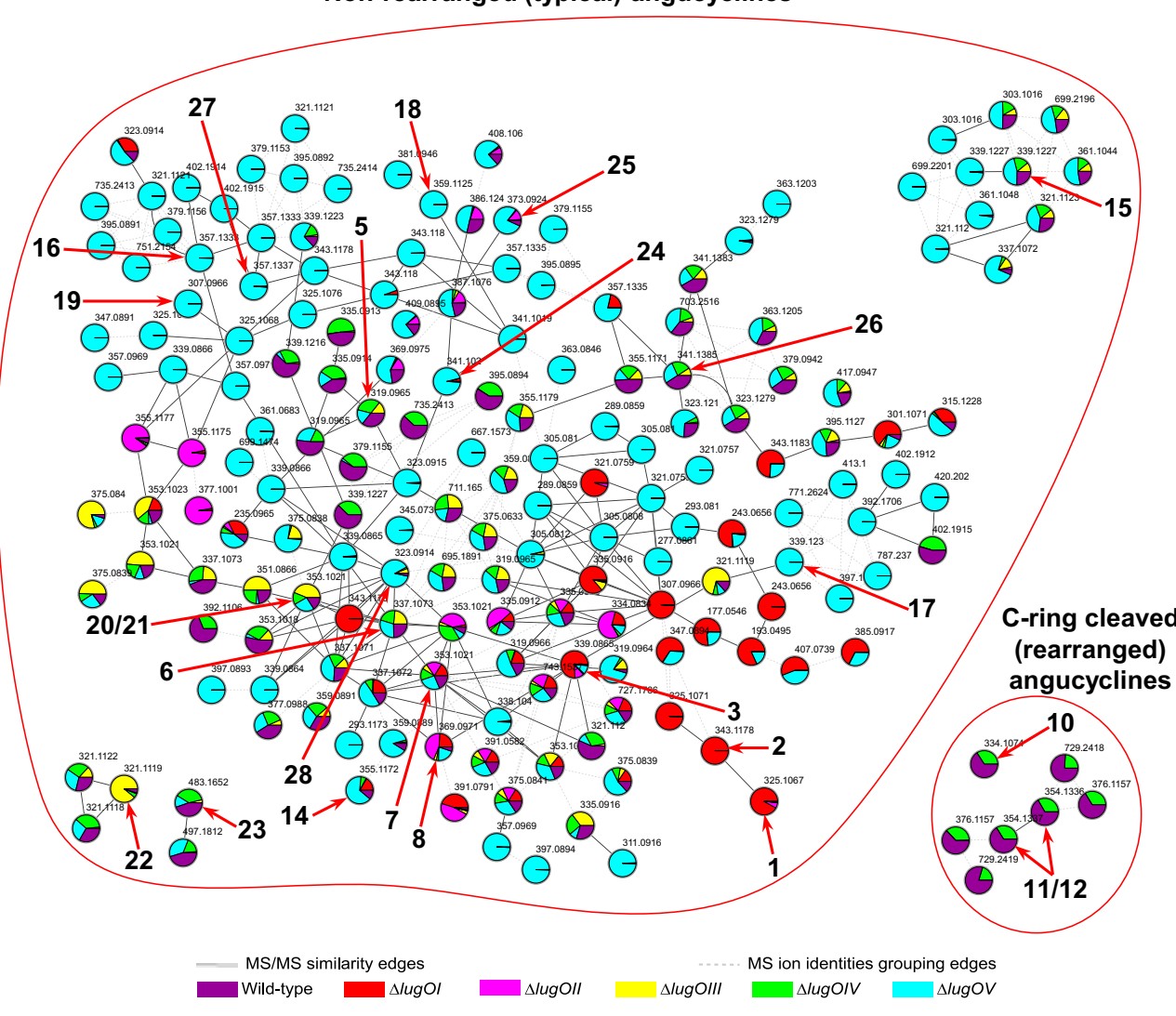

**Fig. 5 Angucycline molecular families identified in the extracts of *Streptomyces* sp. QL37 and its *lugO* mutants, grown on R5 agar.** The nodes are labeled by the precursor mass of their ions and pie charts are mapped to the nodes representing the relative intensities of the ions in the different samples. Arrows are used to highlight the metabolites which were identified in the extracts.

either *lugOIII* or *lugOV* (Figs. 4 and 5). The metabolomic profile of the *lugOIII* and *lugOV* mutants revealed that almost all non-rearranged angucyclines were produced normally, including **5**, **6**, **9**, and **15** (Figs. 4 and 5). Neither mutant produced C-ring cleaved derivatives (**4**, **10–13**), which shows that both LugOIII and LugOV catalyze reactions prior to this key event. Notably, the metabolites produced in the *lugOIII* mutant were more or less shared with the *lugOV* mutant and the wild-type strain. However, in the molecular networks, there was an increased number of nodes, particularly in the non-rearranged angucyclines molecular families, which were exclusively detected in the *lugOV* mutant and were absent in the *lugOIII* mutant (Figs. 4 and 5).

Accordingly, we set out to identify the metabolites accumulating in the *lugOV* mutant. A large scale culture of the *lugOV* mutant on R5 agar medium was extracted with EtOAc, and the extract was subjected to a series of chromatographic fractionation until pure compounds were obtained. Some of the purified compounds were very unstable and would rapidly oxidize or isomerize prior to full chemical characterization. In the end, we managed to purify eleven new (**16–26**) and ten known (**5–7**, **15**, and **28–33**) metabolites (Fig. 6), among which the mass peaks of

**16–19** were exclusively detected in the *lugOV* mutant (Fig. 5). Based on molecular networking analysis, the mass peaks of all the new compounds purified, apart from **22** and **23**, clustered within the non-rearranged angucyclines molecular family, which includes the known angucylinones **5–8** (Fig. 5). This indicates that those new compounds retain the typical angular tetracyclic architecture characteristic of angucyclines, and no major re-arrangements should be expected. This is because clustering of masses in the molecular network is due to similarities in their MS/MS fragmentation pattern, which results from similarities in the chemical structure.

A molecular formula of $C_{20}H_{20}O_6$, with 11 degrees of saturation, could be deduced for compound **16** based on HRESIMS. This was consistent with the signals observed in the $^1H$ and $^{13}C$ NMR spectra. Three exchangeable protons could be established through the HSQC spectrum. A COSY spin system comprising three aromatic protons was observed, and this is characteristic of angucyclinones D-ring (Fig. 1). This was supported by the HMBC correlations observed for the aromatic protons (Supplementary Information Fig. S17). The HMBC correlations from H-10 and $H_3$-8MeO to C-8 confirmed the

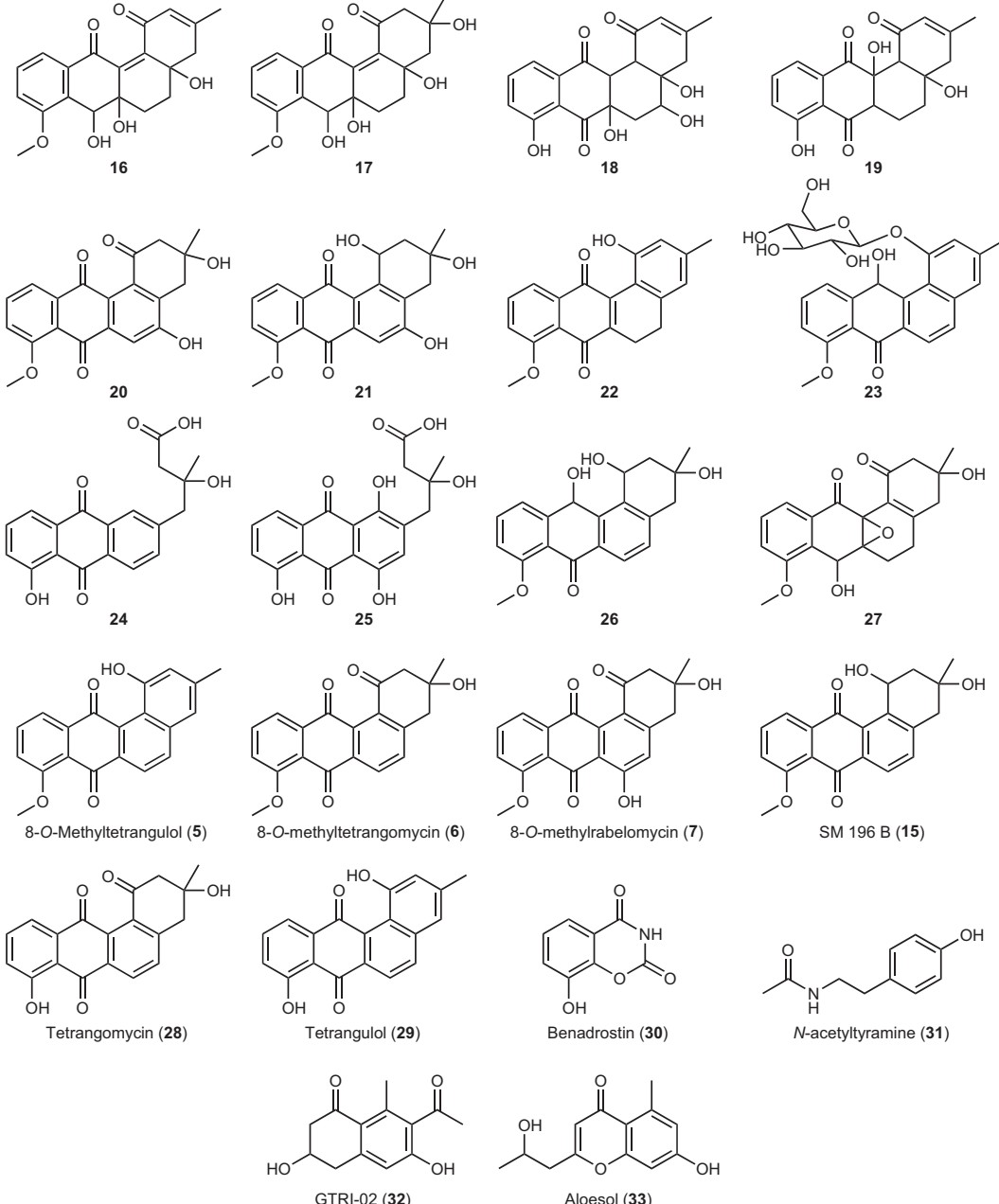

**Fig. 6 Metabolites identified in this study.** Structures of the new (**16–26**) and known (**5–7**, **15**, and **28–33**) metabolites isolated upon the deletion of *lugOV*, and the new metabolite isolated upon its complementation (**27**).

presence of an 8-methoxy substituted angucyclinone. A C-12 keto group was established through the HMBC correlation from H-11 to the carbonyl C-12 ($\delta_c$ 188.2). On the other hand, a hydroxyl group was established at C-7 through the COSY correlation between H-7OH ($\delta_H$ 5.49) and H-7, which in turn has HMBC correlations to C-7a, C-8, and C-11a. The COSY spectrum showed another spin system comprising two aliphatic methylene groups, which were positioned as $CH_2$-5 and $CH_2$-6 based on the HMBC correlation from H-7 to C-6. $CH_2$-6 was connected to CH-7 through the tetrasubstituted oxygenated carbon C-6a ($\delta_c$ 74.1) through the HMBC correlations observed from $H_2$-5, $H_2$-6, and H-7 to C-6a. A hydroxyl group was connected to C-6a through the HMBC correlations from H-6aOH ($\delta_H$ 5.16) to C-6a and C-7. In a similar way the tetrasubstituted oxygenated C-4a ($\delta_c$ 72.4) with its attached hydroxyl group was established. A signal due to the characteristic 3-methyl group of the A-ring in

angucyclinones could be observed in the [1]H NMR spectrum ($\delta_H$ 1.94, 3H, s). Its chemical shift, splitting pattern, and HMBC correlations confirmed its connection to a quaternary olefinic carbon ($\delta_c$ 157.7). An olefinic methine group at C-2 and an aliphatic methylene group at C-4 were established through the HMBC correlations from $H_3$-3Me to C-2 and C-4, from $H_2$-4 to C-4a, and from H-4aOH to C-4. A C-1 keto group was established based on the HMBC correlations from H-2 to C-1 ($\delta_c$ 189.8). The full B-ring including a double bond between C-12a and C-12b, and the connection of C-1 to C-12b were established through the HMBC correlations from H-7 and $H_2$-6 to C-12a ($\delta_c$ 140.7), and from H-2 and H-4aOH to C-12b ($\delta_c$ 148.1). The final connection in the structure was made between the keto group at C-12 and the quaternary olefinic group at C-12a, in agreement with the obtained molecular formula and its required degrees of saturation.

HRESIMS analysis showed that compound **17** has a molecular formula of $C_{20}H_{22}O_7$, with ten degrees of unsaturation. The [1]H NMR spectrum of **17** was similar to that of **16**. The HSQC spectrum showed that in comparison with **16**, compound **17** has four hydroxyl groups instead of three, four aliphatic methylene groups instead of three, and lacks the olefinic methine group. Further analysis of the 2D NMR data (COSY, HSQC and HMBC) revealed that the B-ring to D-ring architecture in **17** was the same as **16** (Supplementary Information Fig. S17). The difference in planar structure was in the A-ring, in which the 3-methyl group in **17** is connected to a tetrasubstituted oxygenated carbon ($\delta_c$ 74.0), which is further connected to a hydroxyl group and two aliphatic methylene groups, positioned at C-2 and C-4. This was established through the HMBC correlations from $H_3$-3Me to C-2, C-3 and C-4, and from H-3OH to C-3 and C-3Me. The C-1 keto group and the connection of the A-ring to the B-ring were established through a similar HMBC correlation pattern to that of **16**.

The molecular formula of **18** is established as $C_{19}H_{18}O_7$ with 11 degrees of unsaturation. 1D and 2D NMR analysis showed the presence of the aromatic D-ring, which has an 8-hydroxy substitution. The A-ring has a similar architecture to that in **16**, apart from C-12b which has an aliphatic methine group. This was evident from the COSY spin system comprising the two CH groups, which are positioned at 12a and 12b, together with the HMBC correlations from H-2 to C-12b and from H-12b to C-1. Another COSY spin system was identified, and this was established to be for an oxygenated CH-5 connected to an aliphatic $CH_2$-6 through the HMBC correlations from $H_2$-4 to C-5 ($\delta_c$ 69.5). Both C-4a and C-6a were tetrasubstituted oxygenated carbons like **16** and **17**. Finally, keto groups were established at both C-7 and C-12 through the HMBC correlations from H-11 to C-12 ($\delta_c$ 194.3), from $H_2$-6 to C-7 ($\delta_c$ 201.1), and from H-12a to both C-7 and C-12.

A molecular formula of $C_{19}H_{18}O_6$, with 11 degrees of unsaturation, was established for **19**. The NMR data of the compound showed the aromatic D-ring with an 8-hydroxy substitution, and an A-ring with C-1 keto group and a 3-methyl on a double bond comprising C-2 and C-3. At position 12b a CH ($\delta_H$ 2.33, s) group was established, which was connected to a tetrasubstituted oxygenated carbon ($\delta_c$ 80.6) at position 12a, through the HMBC correlations from H-12b to C-1 and from H-12aOH to C-12b. In addition, HMBC together with COSY enabled the connection of the tetrasubstituted oxygenated C-4a to the fully reduced $CH_2$-5, followed by $CH_2$-6 then CH-6a. Similar to **18**, compound **19** has both C-7 and C-12 keto groups.

As for compounds **20** and **21**, LC-MS analysis of the purified sample showed a major peak for a compound having a molecular formula of $C_{20}H_{16}O_6$, with 13 degrees of unsaturation. A monoisotopic mass of 337.1072 could also be detected for a less intense coeluting mass peak (Supplementary Data 1 Fig. S53). The [1]H NMR spectrum indeed showed that the sample is a mixture of two compounds at a ratio of around 3:2 (Supplementary Data 1 Fig. S55). Both compounds clearly showed an NMR pattern consistent with the presence of an aromatic D-ring bearing an 8-methoxy substitution. The characteristic HMBC correlation from H-11 to C-12 showed the presence of a keto group at C-12 in both compounds. The resonance signals due to the 3-methyl group in the A-ring were clearly separated for both compounds (Major: $\delta_H$ 1.34, 3H, s; Minor: $\delta_H$ 1.30, 1.8H, s). Their HMBC correlations indicated that both are attached to an oxygenated carbon, which is further attached to two aliphatic methylene groups positioned at C-2 and C-4. An aromatic singlet [1]H signal was in both compounds and this was established to be CH-6 due to the HMBC correlation from $H_2$-4 and H-6 to C-4a, together with an HMBC correlation from H-6 to a ketocarbonyl

carbon (Major: $\delta_c$ 180.3, Minor: $\delta_c$ 181.3) that was positioned as C-7 due to the weak HMBC correlation that is usually observed from H-11 to C-7. An oxygenated aromatic C-5 in both compounds was identified through the HMBC correlation from H2-4 to C-5 (Major: $\delta_c$ 159.8, Minor: $\delta_c$ 160.4). A fully aromatic B-ring could be deduced from the HMBC correlations from $H_2$-2 to C-12b and from H-6 to C-4a and C-12a, in addition to the correlation from $H_2$-4 to C-5. The difference between the two compounds was at C-1, where a keto group was present in the major compound, while a keto-reduced C-1 was identified in the minor compound. This was evident through the HMBC correlations observed for $H_2$-2 in both compounds, together with the COSY spin system comprising $H_2$-2, H-1 and its attached hydroxyl group in the minor compound. The amount of material was not enough to acquire a [13]C NMR spectrum. However, all the carbons of **20** (major) and **21** (minor) could be assigned, apart from C-6a that was tetrasubstituted and did not have any protons within the HMBC correlation distance. The structure elucidated for **20** was consistent with the molecular formula obtained through HRESIMS. As for **21**, its established structure was consistent with the observed mass peak in HRESIMS, which is due to a fragment ion resulting from a loss of a water molecule.

HRESIMS analysis revealed that compound **22** has a molecular formula of $C_{20}H_{16}O_4$. This indicated a reduced form of the previously identified angucyclinone 8-O-methyltetrangulol (**5**, $C_{20}H_{14}O_4$) (Supplementary Data 1 Figs. S59 and S98). The [1]H NMR spectrum of **22** indeed showed a pure compound with resonance signals very similar to **5** (Supplementary Data 1 Fig. S61). The aromatic D-ring protons with an attached 8-methoxy group were detected ($\delta_H$ 7.87, 1H, d: 8.0; 7.69, 1H, t: 8.0; 7.33, 1H, d: 8.0; 4.03, 3H, s). In addition, signals due to an aromatic A-ring with attached 1-hydroxy and 3-methyl substitutions were detected ($\delta_H$ 6.77, 1H, s; 6.65, 1H, s; 9.27, 1H, br s; 2.81, 3H, s). The only difference between the two spectra is the disappearance of the two doublet signals due to the two aromatic protons of CH-5 and CH-6 in **5**, and the appearance of signals due to two aliphatic methylene groups in **22** instead ($\delta_H$ 2.80, 2H, dd: 8.6, 6.4; $\delta_H$ 2.70, 2H, dd: 8.6, 6.4). The splitting pattern of the signals due to the methylene groups indicates that they are connected to each other, which would then position them at C-5 and C-6. The established structure was in perfect agreement with the obtained molecular formula. Re-measurement of **22** for the acquisition of 2D NMR spectra revealed that it is completely oxidized to **5** (Supplementary Data 1 Fig. S61).

The molecular formula of **23** was established as $C_{26}H_{26}O_9$ through HRESIMS analysis. The loss of a hexose sugar was observed during in-source fragmentation and in the MS/MS spectrum. The compound is unstable, because upon re-measuring the purified material for the acquisition of 2D NMR spectra, minor signals due to another compound appeared (Supplementary Data 1 Figs. S64 and S65). The chemical shifts and 2D correlation pattern (Supplementary Information Table S6, Table S8, and Fig. S17) revealed that **23** shares the same angucyclinone as a previously identified α-rhamnose-containing angucycline[22]. As for the sugar moiety, its presence was confirmed through the MS/MS spectrum together with the COSY spin system connecting an anomeric CH proton ($\delta_H$ 5.10) to four oxygenated CH groups and one oxygenated $CH_2$ group (Supplementary Information Fig. S17). The sugar was identified as glucose based on the coupling constant between H-2 and H-3 and between H-3 and H-4, which is 9 Hz for both, indicating an axial orientation of the three protons. Finally, a β conformation was established at the anomeric carbon based on the coupling constant between H-1 and H-2, which is 7.7 Hz.

The purified compound **24** has a molecular formula of $C_{19}H_{16}O_6$, with 12 degrees of unsaturation. 1D and 2D NMR spectra showed the typical pattern of the aromatic D-ring with an attached 8-hydroxy substitution. A C-12 keto group was established through the HMBC correlation from H-11 to C-12. The 3-methyl group in the A-ring could also be observed, and the HMBC correlations from it established the presence of an oxygenated C-3 and two aliphatic methylene groups at C-2 and C-4. Interestingly, an acid/ester carbonyl ($\delta_c$ 172.3) was positioned at C-1 based on the HMBC correlation from $H_2$-2 to C-1. On the other hand, an aromatic CH was positioned at C-12b based on the HMBC correlations from $H_2$-4 to C-12b and from H-12b to C-12. This indicated the cleavage of the A-ring at the bond between C-1 and C-12b, resulting in an acid group at C-1. Two aromatic CH groups were established to be C-5 and C-6 based on the COSY correlation between H-5 and H-6, the HMBC correlations from $H_2$-4 and H-12b to C-5, and the HMBC correlations from H-6 to C-4a. Further HMBC correlations from H-5, H-6 and H-12b established the aromatic B-ring, and particularly the correlation from H-6 to C-7 established a keto group at C-7. The final planar structure of **24** obtained from NMR was consistent with the molecular formula and degrees of unsaturation obtained from HRESIMS.

The molecular formula of **25** was established to be $C_{19}H_{16}O_8$, with 12 degrees of unsaturation, which is two oxygen atoms more than **24**. NMR data analysis indeed revealed that **25** has a similar architecture to **24**, with two hydroxy substitutions at C-6 and C-12b. This was confirmed through the HMBC correlations from H-6OH to C-5 and C-6 ($\delta_c$ 155.8), and the HMBC correlations from $H_2$-4 to C-5 and C-12b ($\delta_c$ 157.2). In addition, the chemical shifts observed for **25** were very similar to those of its previously identified amide analog fridamycin F[46].

Finally, HRESIMS analysis showed that **26** has a molecular formula of $C_{20}H_{20}O_5$, with 11 degrees of unsaturation. The compound is unstable as evidenced during its remeasurement for [13]C NMR analysis (Supplementary Data 1 Figs. S85 and S86). Detailed NMR analysis revealed that **26** is a reduced form of the known metabolite 8-O-methyltetrangomycin (**6**)[26]. The reduction was established to be in the keto groups at C-1 and C-12. Multiple HMBC correlations support a C-1 and C-12 keto reduced structure (Supplementary Information Fig. S17), and the most important were the correlations from H-11 to C-12 and from $H_2$-2 to C-1. A similar structure bearing a C-1 and C-7 ketoreduction was previously reported, and was shown to be prone to autooxidation[47].

Interestingly, a glycoside (**23**) could be identified among the new angucycline-related structures, even though no glycosyl-transferase could be annotated in the *lug* BGC. The same aglycone was previously identified with the attached sugar being either β-glucuronic acid[48] or α-rhamnose[22], while in *Streptomyces* sp. QL37 the sugar was identified to be a β-glucose, showcasing the structural diversity angucyclines can offer. A change in the structure of the attached sugar can have a profound effect on the biological activity of the parent compound[49]. The promiscuity of glycosyltransferases towards either the aglycone or the sugar donors has precedents in natural products biosynthesis[50,51]. As with many streptomycetes, the genome of *Streptomyces* sp. QL37 harbors a multitude of BGCs. It is likely that a glycosyltransferase encoded by another BGC can also accept the angucyclinone as a substrate. Typically, the sugar is initially bound to either a nucleotide di- or monophosphate group before being transferred to the acceptor aglycone by the glycosyltransferase[52].

Importantly, the structures of four of the metabolites (**16–19**) (Fig. 6) indicated that a C6a–C12a epoxide group, which is further reduced or hydrolyzed in the *lugOV* mutant to shunt products, is formed during the biosynthesis. A molecule with an intact epoxide group could not be readily isolated from the *lugOV* mutant, suggesting that LugOV likely plays a role in stabilizing the epoxide moiety. To analyze this further, a construct was generated wherein *lugOV* was positioned behind the constitutive *ermE** promoter in pWHM3-*oriT* and introduced to the *lugOV* null mutant. The genetic complementation restored the ability of the *lugOV* mutant to produce rearranged angucyclinones (**4** and **10–13**), and suppressed the production of **16–19**, which have an opened epoxide ring (Supplementary Information Figs. S18–S22). Interestingly, the LC-MS chromatograms showed a relatively minor peak in the *lugOV* mutant that considerably increased in intensity upon complementation with *lugOV* (Supplementary Information Fig. S18). This peak was then purified from Δ*lugOV::lugOV* as compound **27**, which has a molecular formula of $C_{20}H_{20}O_6$, with 11 degrees of unsaturation. The chemical shifts observed, together with the 2D NMR correlations established the structure of **27** to be the 5,6-dihydro derivative of the previously reported 6a,12a-epoxy containing angucyclinone EI-1507-1[53]. The difference between the two compounds is the two aliphatic methylene groups in **27**, which are positioned at C-5 and C-6 based on the COSY correlations between them, the HMBC correlations from $H_2$-5 and $H_2$-6 to the epoxide carbon C-6a, and the HMBC correlations from $H_2$-5 to the quaternary olefinic carbons C-4a and C-12b. The isolation and identification of the new angucyclinone **27** bearing an intact C6a–C12a epoxide group (Fig. 6) from Δ*lugOV::lugOV* corroborates the role of LugOV in stabilizing the epoxide group.

## Mechanistic model for C-ring cleavage by LugOIII and LugOV.

Taken together, our experiments allow us to propose a detailed biosynthetic pathway for the formation of C-ring cleaved angu-cyclines (Fig. 7). Molecular network analysis revealed that deletion mutants lacking *lugOIII* failed to progress in the biosynthesis beyond the C6 ketoreduction step catalyzed by LugOII. We propose that the next step C6a-C12a epoxidation is catalyzed by LugOIII, which is a new function for ABM monooxygenases and the first discovery of an angucycline epoxidase. Angucycline epoxides have been isolated previously[53], but their associated epoxidases were never identified. A recent study on the *tac* BGC, which is highly similar to *lug*, concluded that the homologs of *lugOIII* (*tacS*) and *lugOV* (*tacT*) are involved in the epoxidation of an angucycline intermediate; however, it was not possible to identify which of the two is the epoxidase[26]. In the same study, the production of C-ring cleaved angucyclines was also abolished upon the deletion of either *tacS* or *tacT*.

In addition to ring-opened epoxide substitution, many of the angucyclinones accumulating in the *lugOV* mutant have an 8-hydroxy substitution rather than the 8-methoxy derivatives, which were predominant in the wild-type (Figs. 5 and 6). This suggests that the next step is catalyzed by the C8 O-methyltransferase LugN and LugOV acts preferentially on 8-methoxy-containing intermediates. The molecular network data further indicates that LugOIII can act on both methylated and non-methylated intermediates. Indeed, the accumulation of the 8-hydroxy analogs was suppressed upon the complementation of *lugOV* in the *lugOV* mutant (Supplementary Information Fig. S18). It is worth mentioning that almost all of the C-ring cleaved angucyclinones identified so far have an 8-methoxy substitution[25]. Conversely, the 8-methoxy substitution was found to block B-ring cleavage[15].

LugOV is likely the ultimate post-PKS enzyme directing the biosynthetic pathway towards the C-ring cleaved metabolites. The exact molecular mechanism by which LugOV contributes to

**Fig. 7 Proposed post-PKS modifications leading to C-ring cleavage and rearrangement of angucyclinones.** Blue arrows represent shunt pathways that diversify the outcome of the *lug* BGC.

C-ring cleavage is yet to be resolved. A previous study on the biosynthesis of the hatomarubigin angucyclines proposed that HrbF, a direct homolog of LugOV, regulates the regiospecificity of oxygenation enzymes; this idea was based on the detection of the 5-hydroxy substituted angucyclinone, hatomarubigin F, only in the absence of *hrbF*[54]. However, our mass spectrometry data clearly show that both 5-hydroxy-containing compounds **20** and **21** are produced in the presence of *lugOV* (Fig. 5). This indicates that it is unlikely for LugOV, or its homolog HrbF, to have a direct role in the regiospecificity of the angucyclinones oxygenases. The discrepancy is likely explained by the higher sensitivity of our mass spectral data as compared to UV detection used for the detection of the hatomarubigin metabolites.

We propose a pathway for angucyclinones C-ring cleavage leading to lugdunomycin (**4**) wherein two routes are likely to proceed (Fig. 7). Path A is based on a Grob-type fragmentation, while path B is based on the BVO mechanism. Either mechanism requires several reduction steps for their completion, particularly at C7, which warrants the future study of the reductases encoded in the *lug* cluster. Importantly, epoxidation is an essential prior step to both pathways, and not an alternative route to BVO as generally believed[25,55].

To the best of our knowledge, Grob-type fragmentation has so far not been experimentally proven as a mechanism for type II post-PKS enzymes. A recently published review on C-ring cleaved angucyclinones proposed three mechanisms for C-ring cleavage, including Grob-type fragmentation, which all proceed through the fragmentation of an epoxide intermediate[25]. However, for the formation of almost all of the C-ring cleaved angucyclinones, the proposed mechanisms require an epoxide intermediate (**34**) in which both keto groups at C7 and C12 have been reduced (Supplementary Information Fig. S23). This was elaborated for intermediate **35** which is crucial in the pathway towards the biosynthesis of **4**[23]. At the same time, the enzymes responsible for the ketoreduction of C7 and C12 have been identified through targeted deletion of their encoding genes *tacO* and *tacA*, respectively[26]. During this study, compound **35** could still be produced in the *tacA* mutant, indicating that it is biosynthesized from an intermediate having only a C7 ketoreduction, as per our proposed mechanism.

An epoxidation followed by a BVO reaction was earlier proposed as the mechanism of the monooxygenase XanO4 for the cleavage and rearrangement of an anthraquinone intermediate, resulting in the formation of a xanthone ring in the type II PKS-derived natural product xantholipin[31]. Interestingly, in vitro reactions utilizing the monooxygenase FlsO1 resulted in a C-ring cleavage and production of a xanthone compound from prejadomycin through the same mechanism[56]. FlsO1 is distantly related to LugOIII and LugOV, and acts in vivo as a C5 hydroxylase following the B-ring cleavage enzyme FlsG, in the biosynthesis of fluostatins[57]. In addition, based on our data the epoxidation step is not catalyzed by the C-ring cleavage enzyme required for lugdunomycin (**4**) biosynthesis, making LugOV then biochemically unique as compared to XanO4 or FlsO1. Thus, in vitro assays utilizing the purified enzymes, which have already initiated in our lab, would help reveal the exact mechanism by which the novel enzyme LugOV, either directly or indirectly, catalyzes C-ring cleavage in angucyclinones.

## Conclusion

Our work is the first to be aimed at studying the enzymes and likely mechanisms involved in the C-ring cleavage of angucyclinones. We focused on investigating the putative oxygenases in the *lug* BGC that was identified to govern the production of multiple non-rearranged as well as C-ring cleaved rearranged angucyclinones. Data analysis revealed that out of five studied genes, four have a hierarchical role towards C-ring cleavage. They start with LugOI, which performs C12 hydroxylation, a base post-PKS tailoring reaction from which the biosynthetic route to several angucycline families diverge. LugOI is proposed to have an added role in preventing the tautomerization of the C6 keto group of the generated intermediate, enabling its reduction by LugOII. Unlike B-ring cleavage, the C6 ketoreduction performed by LugOII, together with the 8-methoxy substitution, is essential for C-ring cleavage. The downstream oxygenase following LugOII was found to be LugOIII which was identified for the first time as the epoxidase catalyzing the installation of a C6a–C12a epoxide group, a prerequisite for C-ring cleavage. An epoxidation is a new function for antibiotic biosynthesis monooxygenase family of enzymes, to which LugOIII belongs. Ultimately, the novel enzyme LugOV stabilizes the epoxide group and facilitates C-ring cleavage either directly or indirectly, and the exact reaction mechanism is yet to be uncovered.

The current study provides essential preliminary information towards the delineation of the pathway leading to C-ring cleavage in angucyclinones. In addition, it provides important clues for designing future effective in vitro enzymatic reactions, aiming at understanding the production of C-ring cleaved rearranged angucyclinones and increasing their structural diversity.

## Methods

**Bacterial strains and growth conditions**. Bacterial strains used in this study are indicated in Supplementary Information Table S2. *Streptomyces* sp. QL37 was obtained from the soil in the Qinling mountains (P. R. China)[58]. The strain is deposited to the collection of the Centraal Bureau voor Schimmelcultures (CBS) in Utrecht, the Netherlands, under deposit number 138593. *Escherichia coli* JM109 was used for general cloning and was grown on Luria Broth (LB) without antibiotics. *E. coli* ET12567/pUZ8002 was used for conjugation of plasmids to *Streptomyces* sp. QL37[40,59,60]. Strains containing plasmids were selected on LB containing ampicillin (final $100 \mu g \, mL^{-1}$), apramycin ($50 \mu g \, mL^{-1}$), chloramphenicol ($25 \mu g \, mL^{-1}$) and kanamycin ($50 \mu g \, mL^{-1}$). Spore stocks were prepared according to Kieser et al.[39] from cultures of *Streptomyces* sp. QL37 grown on SFM agar for seven days at 30 °C. The spore stocks were stored in 20% glycerol at −20 °C.

**Construction of knock-out mutants**. For construction of the knock-out mutants the plasmid pWHM3-*oriT* was used, a derivative of the plasmid pWHM3, which harbors the *oriT* in the NdeI site, allowing its conjugative transfer[23,61,62]. The in-frame deletion mutants were created according to a method previously described[37]. Using this method, a knock-out construct was generated containing an apramycin cassette that is flanked by an up- and downstream region of the targeted gene. The ~1.5 kb upstream and downstream region of the targeted gene were amplified from *Streptomyces* sp. QL37 genomic DNA with their respective primers (Supplementary Information Table S3). These were subsequently cloned in pWHM3-*oriT* using the correct restriction enzymes. An apramycin resistance cassette flanked by two *loxP* sites was cloned in between the fragments using XbaI[41]. The integrity of the construct was verified using Sanger sequencing and restriction enzyme analysis. The plasmid was transformed in the methylase deficient strain ET12567/pUZ8002 that allows conjugation of the plasmid to *Streptomyces* sp. QL37. The conjugation was executed as described by Kieser et al.[39]. However, for conjugation *Streptomyces* sp. QL37 was grown on SFM containing 60 mM $MgCl_2$ and 60 mM $CaCl_2$[60]. The correct mutants were selected by their resistance against apramycin

(50 µg mL$^{-1}$) and sensitivity to thiostrepton (10 µg mL$^{-1}$). In order to remove the apramycin cassette the pUWL-Cre construct was conjugated to the apramycin cassette containing strains[41]. The removal of the apramycin cassette was verified by polymerase chain reaction (PCR) with the primers annotated in Supplementary Information Table S3 and Sanger sequencing of the PCR products[37].

**Generation of constructs for the complementation of *lugOI*, *lugOIII*, and *lugOV* mutants**. To verify whether LugOI has a similar function as PgaE[34] in angucycline biosynthesis, a construct for constitutive expression of *pgaE* was generated. To do so, *pgaE* was cloned under the control of the SP44 promoter in the pS-GK plasmid, lacking the kanamycin reporter gene[63]. This plasmid comprises the T7 terminator, the SP44 promoter, the ribosomal binding site (RBS-SR41) and the reporter gene, sf*gfp* (super folder green fluorescent protein), cloned into pSET152. The *pgaE* gene was amplified from the genome of *Streptomyces* sp. PGA64 as described by Palmu et al.[64]. The *pgaE* fragment was then cloned in pS-GK using the SpeI and EcoRV restriction enzymes, resulting in the final construct pS-GK-*pgaE*.

To complement the *lugOIII* and *lugOV* mutants, constructs for constitutive expression of *lugOIII* and *lugOV* were generated. Therefore, *lugOIII* and *lugOV* were amplified from the genome of *Streptomyces* sp. QL37 using the primers indicated in Supplementary Information Table S3 and were cloned downstream the *ermE** promoter. The PCR product of *lugOIII* was digested with NdeI and XbaI and the plasmid pWHM3-*oriT* was linearized using EcoRI and XbaI. A three-way ligation including the *ermE** promoter, pWHM3-*oriT* and *lugOIII* resulted in the final construct for the complementation of the *lugOIII* mutant. For the construction of the plasmid for the constitutive expression of *lugOV*, the PCR product of *lugOV* was digested with NdeI and BamHI and the plasmid pWHM3-*oriT* was linearized using EcoRI and BamHI. A three-way ligation using the latter two fragments and *ermE** resulted in the final construct for complementation of the *lugOV* mutant.

The integrity of the constructs was verified using Sanger sequencing and restriction enzyme analysis. The constructs were isolated from *E. coli* JM109 and transformed into the methylase deficient strains *E. coli* ET12567/pUZ8002 that was used to conjugate the plasmids towards either the *lugOI*, *lugOIII* or *lugOV* mutant[40,59,60]. Ex-conjugants carrying pS-GK-*pgaE* construct were selected on SFM medium containing 50 µg mL$^{-1}$ apramycin. Ex-conjugants carrying the *lugOIII* or *lugOV* complementation constructs were selected on SFM medium containing 20 µg mL$^{-1}$ thiostrepton in order to maintain the plasmid.

**Bioinformatics**. The amino acid sequence alignment of LugOIII and LugOV was performed using Clustal Omega[65] and visualized using ESPript 3.0[66]. The protein sequences of all the angucycline BGCs of known compounds were picked according to literature. Protein sequences were retrieved from the NCBI database or from MIBiG[67]. A phylogenetic tree was generated using MEGA-X[68]. The sequences were aligned using the MUSCLE algorithm. A Maximum Likelihood tree was generated with a bootstrap value of 500. For the visualization and annotation, the software tool iTol was used[69]. Protein similarity searches were executed using the Basic Local Alignment Search (BLAST) Tool against the NCBI database[70]. GBK files from biosynthetic gene clusters were derived from antiSMASH[71]. The BGCs were drawn using Clinker[72].

**Extraction of natural products**. *Streptomyces* sp. QL37 spores were confluently grown on agar plates containing 25 mL Minimal Medium (MM) supplemented with 0.5% mannitol and 1% glycerol, or R5 (Difco) supplemented with 1% mannitol and 0.8% peptone[23]. The experiment was conducted in triplicate. After seven days of growth at 30 °C the agar plates were cut into small pieces and soaked in 25 mL of ethyl acetate for 12 hours. Subsequently the ethyl acetate was decanted and evaporated at room temperature. This process was repeated two times. The dried extract was re-dissolved in methanol (MeOH) and centrifugated in order to remove any undissolved matters. The obtained MeOH solutions were transferred to new pre-weighed glass vials, where it was dried under nitrogen. The crude extracts were weighed and dissolved in methanol to a final concentration of either 1 mg mL$^{-1}$ (extracts derived from MM medium) or 0.5 mg mL$^{-1}$ (extracts derived from R5 medium). The prepared solutions were centrifuged again for 20 min at 4 °C in order to remove any suspended matters.

**LC-MS/MS analysis**. LC-MS/MS acquisition was performed using Shimadzu Nexera X2 UHPLC system, with attached PDA, coupled to Shimadzu 9030 QTOF mass spectrometer, equipped with a standard ESI source unit, in which a calibrant delivery system (CDS) is installed. The dry extracts were dissolved in MeOH to a final concentration of 1 mg mL$^{-1}$ or 0.5 mg mL$^{-1}$, and 2 µL were injected into a Waters Acquity HSS C$_{18}$ column (1.8 µm, 100 Å, 2.1 × 100 mm). The column was maintained at 30 °C, and run at a flow rate of 0.5 mL min$^{-1}$, using 0.1% formic acid in H$_2$O as solvent A, and 0.1% formic acid in acetonitrile as solvent B. A gradient was employed for chromatographic separation starting at 5% B for 1 min, then 5–85% B for 9 min, 85–100% B for 1 min, and finally held at 100% B for 3 min. The column was re-equilibrated to 5% B for 3 min before the next run was started.

All the samples were analyzed in positive polarity, using data dependent acquisition mode. In this regard, full scan MS spectra (*m/z* 100–1700, scan rate 10 Hz, ID enabled) were followed by two data dependent MS/MS spectra (*m/z* 100–1700, scan rate 10 Hz, ID disabled) for the two most intense ions per scan. The ions were selected when they reach an intensity threshold of 1500, isolated at the tuning file Q1 resolution, fragmented using collision induced dissociation (CID) with fixed collision energy (CE 20 eV), and excluded for 1 s before being re-selected for fragmentation. The parameters used for the ESI source were: interface voltage 4 kV, interface temperature 300 °C, nebulizing gas flow 3 L min$^{-1}$, and drying gas flow 10 L min$^{-1}$.

A multiple reaction monitoring (MRM) MS method was used to target the detection of compounds **11** and **12** at higher sensitivity. The MRM transition used was 354.1336 → 318.1125, with an *m/z* range of ± 20 ppm, at a scan rate of 5 Hz, a fixed collision energy of 20 eV, and ID function disabled. All other LC and MS settings are as given above.

The MS system was tuned using standard NaI solution (Shimadzu). The same solution was used to calibrate the system before starting. In addition, a calibrant solution made from ESI tuning mix (Sigma-Aldrich) was introduced through the CDS system, the first 0.5 min of each run, and the masses detected were used for post-run mass correction for the file, ensuring stable accurate mass measurements. System suitability was checked by including a standard sample made of 5 µg mL$^{-1}$ paracetamol, reserpine, and sodium dodecyl sulfate; which was analyzed regularly in between the batch of samples.

**Analysis of the angucycline standards**. Compounds **1**–**3** were previously isolated in other studies[4,73]. Compounds **4**–**12** have previously been isolated and characterized from the wild-type *Streptomyces* sp. QL37[23]. Chemically synthesized compound **13**[74]

was kindly provided by M.T. Uiterweerd and Prof. Dr. A.J. Minnaard. Compounds **14** and **15** were previously produced through in vitro enzymatic reactions utilizing purified LugOIIred[38]. All those compounds, apart from compound **9**, were re-analyzed using LC-MS and were identified in the crude extract of wild-type *Streptomyces* sp. QL37 culture by matching their corresponding peaks using their retention time, UV, MS, and MS/MS spectra. Compound **9** is very unstable and was already oxidized upon re-analysis. However, its corresponding peak was annotated in the crude extract of the MM cultures, from which it was previously isolated, based on its exact mass and similarity of the UV spectrum to its previously reported one[22].

**Molecular networking**. For molecular networking the Global Natural Products Social Molecular Networking (GNPS) web tool was used[43]. Within the GNPS, the Feature Based Molecular Networking (FBMN) platform was used in combination with Ion Identity Molecular Networking (IIMN)[75,76]. Initially, the raw LC-MS/MS data were converted to mzXML format using LabSolutions (Shimadzu) (MSV000091727). The converted files were processed using MZmine 2.53 for mass detection, chromatographic peak deconvolution, and alignment[42]. The data were further processed in MZmine 2.37.1.IIN_17.7_LS to detect the different ion species of the same molecule for the generation of the IIMN. The parameters used for data processing are summarized in Supplementary Information Table S4. The processed data were subsequently exported to GNPS using the default settings. The generated peak lists were filtered in Excel to remove the mass features which are present in the media blanks and *lugA–C* mutant. In that regard mass features were retained if their average peak area in any sample is higher than 3000 and at least 50 fold higher than in the media or the *lugA–C* mutant. Finally, the data were submitted to FBMN in GNPS for the generation of the molecular networks using the default settings. The networks were visualized using Cytoscape[77]. The molecular networking jobs can be accessed at https://gnps.ucsd.edu/ProteoSAFe/status.jsp?task=3bb61b8abb3f4356ae9ab7ed1330b785 and https://gnps.ucsd.edu/ProteoSAFe/status.jsp?task=ab35f43720344b748de2b92d35cd3109, for extracts of MM and R5 grown cultures, respectively.

**Isolation and characterization of angucyclines**. The strains *Streptomyces* sp. QL37 Δ*lugOV* and Δ*lugOV::lugOV* were grown to isolate the new angucycline masses accumulating in them. Briefly, large scale cultures of Δ*lugOV* (5.25 L) and Δ*lugOV::lugOV* (7.5 L) were prepared on R5 agar medium supplemented with 1% mannitol and 0.8% peptone. The cultures were incubated at 30 °C for seven days, and then the agar was cut into small pieces and extracted three times with ethyl acetate overnight at room temperature. The extracts were evaporated under reduced pressure, and the resulting material was adsorbed on silica gel and loaded on vacuum liquid chromatography (VLC) column (125 × 50 mm, silica gel 60 Å pore size, 230–400 mesh, Sigma). After defatting with n-hexane, a gradient elution was employed for the column using different ratios of ethyl acetate and methanol. A second VLC column (30 × 300 mm) was performed for one of the fractions obtained from the Δ*lugOV* extract, and was eluted with a gradient mixture of dichloromethane:methanol.

All fractions obtained were analyzed using LC-MS to monitor the fractionation process and guide the further purification steps. Fractions containing the peaks of interest were subjected to (repeated) HPLC purification using gradients of water:methanol mixtures until pure compounds were obtained. The HPLC purifications were performed on a Waters preparative HPLC

system equipped with a photodiode array detector (PDA). The columns used were SunFire $C_{18}$ column (10 μm, 100 Å, 19 × 150 mm) or SunFire $C_{18}$ column (5 μm, 100 Å, 10 × 250 mm). The isolated compounds were first checked for purity and novelty using [1]H NMR spectrum, and new compounds with good purity levels were analyzed with 2D NMR. The NMR data were recorded on Bruker AVIII-600 or Ascend 850 NMR spectrometer. NMR spectra were analyzed using the software MestReNova.

Compound **16**: yellowish solid; yield: 0.005 g (0.01%); [1]H and [13]C NMR data see Supplementary Information Table S5 and Table S7; HRMS (ESI+) *m/z*: $[M + H]^+$ calcd for $C_{20}H_{21}O_6^+$ 357.1333, found 357.1331 $[M + H\text{-}2H_2O]^+$ calcd for $C_{20}H_{17}O_4^+$ 321.1121, found 321.1121; UV $\lambda_{max}$ (LC-MS): 207, 285, 350(sh) nm.

Compound **17**: yellowish solid; yield 0.006 g (0.01%); [1]H and [13]C NMR data see Supplementary Information Table S5 and Table S7; HRMS (ESI+) *m/z*: $[M + H\text{-}2H_2O]^+$ calcd for $C_{20}H_{19}O_5^+$ 339.1227, found 339.1233 $[2M + NH_4]^+$ calcd $C_{40}H_{48}NO_{14}^+$ 766.3069, found 766.3069; UV $\lambda_{max}$ (LC-MS): 200, 261, 284, 345(sh) nm.

Compound **18**: yellowish solid; yield 0.002 g (0.005%); [1]H and [13]C NMR data see Supplementary Information Table S5 and Table S7; HRMS (ESI+) *m/z*: $[M + H]^+$ calcd for $C_{19}H_{19}O_7^+$ 359.1125, found 359.1126 $[M+Na]^+$ calcd for $C_{19}H_{18}NaO_7^+$ 381.0945, found 381.0942; UV $\lambda_{max}$ (LC-MS): 229, 272, 346 nm.

Compound **19**: dark yellow solid; yield 0.005 g (0.01%); [1]H and [13]C NMR data see Supplementary Information Table S5 and Table S7; HRMS (ESI+) *m/z*: $[M + H]^+$ calcd for $C_{19}H_{19}O_6^+$ 343.1176, found 343.1176 $[M + H\text{-}2H_2O]^+$ calcd for $C_{19}H_{15}O_4^+$ 307.0965, found 307.0968; UV $\lambda_{max}$ (LC-MS): 232, 350 nm.

Compound **20**: dark yellow solid; yield 0.003 g (0.006%); [1]H and [13]C NMR data see Supplementary Information Table S5 and Table S7; HRMS (ESI+) *m/z*: $[M + H]^+$ calcd for $C_{20}H_{17}O_6^+$ 353.1020, found 353.1019 $[M+Na]^+$ calcd $C_{20}H_{16}NaO_6^+$ 375.0839, found 375.0839; UV $\lambda_{max}$ (LC-MS): 218, 278, 379 nm.

Compound **21**: dark yellow solid; yield 0.002 g (0.004%); [1]H and [13]C NMR data see Supplementary Information Table S5 and Table S7; HRMS (ESI+) *m/z*: $[M + H\text{-}H_2O]^+$ calcd for $C_{20}H_{17}O_5^+$ 337.1071, found 337.1072 $[M + H\text{-}2H_2O]^+$ calcd for $C_{20}H_{15}O_4^+$ 319.0965, found 319.0966; UV $\lambda_{max}$ (LC-MS): 218, 278, 379 nm.

Compound **22**: brownish solid; yield 0.019 g (0.041%); [1]H NMR spectrum see Supplementary Data 1 Fig. S61; HRMS (ESI+) *m/z*: $[M + H]^+$ calcd for $C_{20}H_{17}O_4^+$ 321.1121, found 321.1125; UV $\lambda_{max}$ (LC-MS): 212, 267, 289, 413 nm.

Compound **23**: yellowish solid; yield 0.001 g (0.001%); [1]H and [13]C NMR data see Supplementary Information Table S6 and Table S8; HRMS (ESI+) *m/z*: $[M + H]^+$ calcd for $C_{26}H_{27}O_9^+$ 483.1650, found 483.1651 $[M + H\text{-}Glc]^+$ calcd for $C_{20}H_{17}O_4^+$ 321.1121, found 321.1121; UV $\lambda_{max}$ (LC-MS): 220, 280, 344 nm.

Compound **24**: dark yellow solid; yield 0.021 g (0.045%); [1]H and [13]C NMR data see Supplementary Information Table S6 and Table S8; HRMS (ESI+) *m/z*: $[M + H]^+$ calcd for $C_{19}H_{17}O_6^+$ 341.1020, found 341.1020 $[M + H\text{-}H_2O]^+$ calcd for $C_{19}H_{15}O_5^+$ 323.0914, found 323.0916; UV $\lambda_{max}$ (LC-MS): 217, 259, 282, 314, 402 nm.

Compound **25**: reddish solid; yield 0.002 g (0.003%); [1]H and [13]C NMR data see Supplementary Information Table S6 and Table S8; HRMS (ESI+) *m/z*: $[M + H]^+$ calcd for $C_{19}H_{17}O_8^+$ 373.0918, found 373.0916 $[M + H\text{-}H_2O]^+$ calcd for $C_{19}H_{15}O_7^+$ 355.0812, found 355.0811; UV $\lambda_{max}$ (LC-MS): 223, 293, 483 nm.

Compound **26**: dark yellow solid; yield 0.01 g (0.022%); [1]H and [13]C NMR data see Supplementary Information Table S6 and

Table S8; HRMS (ESI+) $m/z$: $[M + H]^+$ calcd for $C_{20}H_{21}O_5^+$ 341.1384, found 341.1386 $[M + H\text{-}H_2O]^+$ calcd for $C_{20}H_{19}O_4^+$ 323.1278, found 323.1278; UV $\lambda_{max}$ (LC-MS): 200, 216, 265, 294, 334 nm.

Compound **27**: colorless solid; yield 0.025 g (0.318%); $^1H$ and $^{13}C$ NMR data see Supplementary Information Table S6 and Table S8; HRMS (ESI+) $m/z$: $[M + H]^+$ calcd for $C_{20}H_{21}O_6^+$ 357.1333, found 357.1337 $[M+Na]^+$ calcd for $C_{20}H_{20}NaO_6^+$ 379.1152, found 379.1154; UV $\lambda_{max}$ (LC-MS): 217, 260(sh), 314 nm.

**Reporting summary**. Further information on research design is available in the Nature Portfolio Reporting Summary linked to this article.

## Data availability
LC-MS/MS data are publicly available on MassIVE repository (MSV000091727). Copies of the spectra of the metabolites identified in this study (HRMS, UV, and NMR spectra) are provided in the Supplementary Data 1 file, and the raw data are available upon request.

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

## Acknowledgements

We thank the Dutch Research Council (NWO) NACTAR programme (project number 16439 to G.P.vW.) and the Novo Nordisk Foundation (project number NNF19OC0057511 to M.M.-K.) for the financial support. We thank the NMR facility, Leiden University, the Netherlands, for help with NMR measurements. The NMR experiments performed in this work were supported partly by uNMR-NL Grid, a distributed, state-of-the-art Magnetic Resonance facility for the Netherlands (NWO grant 184.035.002).

## Author contributions

Conceptualization, G.P.vW.; Methodology, S.S.E. and H.U.vdH.; Formal analysis, S.S.E. and H.U.vdH.; Investigation, H.U.vdH., X.X., A.N., L.R.B., and C.W.; Writing – Original draft, S.S.E., and H.U.vdH.; Writing – Review & editing, S.S.E., M.M.-K. and G.P.vW.; Visualization, S.S.E. and H.U.vdH.; Supervision, S.S.E., M.M.-K., and G.P.vW.; Funding acquisition, G.P.vW.

## Competing interests

The authors declare no competing interests
