## [Peer Review File · Communications Chemistry]

Reviewers' comments:

Reviewer #1 (Remarks to the Author):

Elsayed and co-workers report new genetic and biosynthetic studies on a previously uninvestigated class of angucycline-type oxygenases responsible for cleaving the C ring. Lugdunomycin is an unusual antibiotic that is hypothesized to derive from several unusual oxidative rearrangements. As a result, this manuscript reports new enzymology and begins to unravel the unique oxidative cascade that is responsible for the biosynthesis of the lugdunomycin skeleton. The angucyclines are also highly significant for their plethora of biological activities, which makes this manuscript both very timely and impactful in terms of its results. The manuscript is very well-written and will be of great interest to the community and wider field.

First, the authors carried out phylogenetic analyses of the lugdunomycin oxygenase genes to evaluate their position within the greater context of other oxygenases that are implicated in oxidative ring cleavage. The introduction and phylogenetic analyses are especially helpful for establishing the evolutionary heritage of the pathway.

Secondly, The authors carried out targeted deletion of oxygenases lugOI, lugOII, lugOIII, lugOIV, and lugOV. The authors also carried out genetic complementation studies, either with the lug oxygenase or other angucycline oxygenases, which provided evidence as to the function of the gene product and ruled out polar effects due to the knockout. These studies resulted in strains that collectively produced 12 new angucycline metabolites. Each of these metabolites was rigorously characterized via NMR spectroscopy and high-resolution mass spectrometry. Furthermore, the structural analysis of these metabolites provides important insight into the biosynthetic logic of the lugdunomycin oxidative cascade.

Third, and rather illuminatingly, the authors have employed comprehensive metabolome analysis and molecular networking to deconvolute the complex HPLC-MS/MS data that were generated in the study. This is an essential contribution to the manuscript that aids in the interpretation of the complicated structural data that is presented. Or, stated another way, this is an integral contribution that facilitates the interpretation of the biosynthesis of the plethora of metabolites that are accumulated in these strains.

This manuscript is highly significant - it describes a brand new class of C-ring cleaving oxygenases - which should open the door for the study of similar enzymes discovered in many other angucycline gene clusters. I recommend that this manuscript be accepted to Communications Chemistry following minor corrections.

The comments below are minor corrections that should be addressed before the manuscript is accepted for publication:

Page 3, Line 56: The authors introduce the abbreviation "BVO" to refer to Baeyer-Villiger Monooxygenases. Please ensure that this abbreviation is used consistently throughout - Fig. 3 looks like it uses "BVMO" instead of BVO. This should be updated throughout the manuscript.

Page 5, Line 35: The others use the spelling "orthologue" here and "ortholog" elsewhere in the manuscript. The spelling should be corrected throughout the manuscript for consistency.

Page 7, Line 109: Fig. 3 features five labels under the header "Middle Ring" that are not displaying correctly. 'B-ring cleavage oxygenases', 'Type 'O' BVMO oxygenases', 'Type '1' BVMO oxygenases', 'Type 'II' BVMO oxygenases', and 'Anthrone oxygenases' all appear like the text was corrupted during the conversion of the file to .PDF or .EPS. The authors should re-generate the figure to enhance the legibility of the labels.

Page 8, Lines 136 - 144: The authors make a very significant observation regarding the production of limamycins and lugdunomycin on R5 and MM media. This is very interesting that the production of these different metabolites is media-dependent, and it gives additional weight to the metabolomics data that is presented below!

Page 11, Lines 185 - 191: It is quite gratifying that the knock-out of lugOII resulted in the abrogation of C6-ketoreduction and a significant change in the metabolome, as expected!

Page 13, Line 232: Change it's to its

Page 17, Lines 309 - 319, Lines 348-353 : The production of compound 23 with a glucoside is interesting. There do not appear to be any glycosyltransferases or other glucosylated lug metabolites (as mentioned by the authors in Lines 348 - 353). It would be helpful for the reader if the authors could provide some sort of mechanistic interpretation for how this glucoside was formed in the discussion section.

Page 17, Lines 310 - 311: Please correct the syntax here regarding the degradation of 23. It is unclear what is meant by

"because upon the acquisition of the 2D NMR spectra minor signals due to another compound has increased in intensity". Is compound 23 sufficiently pure to obtain good quality NMR spectra? The spectra appear sufficiently clean for this compound in the supporting information.

Page 22, Line 409 - 419: the discussion about Grob-type fragmentation is insightful and would be unique to the lugdunomycin pathway, since this has apparently not been described previously for angucyclines.

Reviewer #2 (Remarks to the Author):

The article by Somayah S. Elsayed, Helga U. van der Heul, Xiansha Xiao, Aleksi Nuutila, Laura R. Baars, Changsheng Wu, Mikko Metsä-Ketelä, Gilles P. van Wezel "Unravelling key enzymatic steps in C-ring cleavage during angucyclines biosynthesis" is significant for angucyclinone biosynthetic pathways clarification and complements recent findings in this field with new experiments. The authors have fully discussed the history of the subject with all the examples and acknowledged the latest hypotheses on the origin of the rearranged angucyclinones. The supplementary file is well-organized and sufficiently detailed for the readers.

I suggest the manuscript can be published in its current form with only one small alteration. The statement that this "study showed for the first time that angucyclinone C-ring cleavage proceeds through epoxidation" is a little bit of a stretch. There are at least two experiment proofs shown in the previous papers (one of them could be cited as well):

1) Fu, X.-Z. et al. / *J. Org. Chem.*, 2022, 87, 23, 15998–16010. (DOI: 10.1021/acs.joc.2c02134)

2) Reference # 26 from the current manuscript (Cao, M. M. et al. / *Angew. Chem. Int. Ed.*, 2021, 60, 7140–7147 (DOI: 10.1002/anie.202015570) states: "Knock-out experiments therein (pages S68-S69) reliably show, that, at least, part of the fragmented angucyclinones (compounds #2,3 in this paper) are formed according to "epoxide hypothesis" and demonstrates the pathway in Figure S30 (https://onlinelibrary.wiley.com/action/downloadSupplement?doi=10.1002/anie.202015570&file=anie202015570-sup-0001-misc_information.pdf).

But this aspect does not diminish the quality of the current manuscript, which is fully suitable for the *Communications Chemistry* journal. The article will be noticeable in its field providing new evidence in the revision of rearranged angucyclinones biosynthesis.

Reviewer #3 (Remarks to the Author):

The study by Elsayed et al. gives insights into the biosynthesis of lugdunomycin by targeted deletion of five oxygenase genes, in combination with molecular networking and structural elucidation of eleven new metabolites. The key finding is the identification of LugOIII as the epoxidase catalyzing the installation of a C6a–C12a epoxide group, which appears to be the prerequisite for C-ring cleavage. In addition, the authors could show that LugOV is also essential for the biosynthesis of C-ring cleaved angucyclines and can stabilize the epoxide group. Towards this end the data is convincing and the conclusions made by the authors are sound. The findings are an advance to the recent finding that the LugOIII/OV homologs TacS/T are involved in epoxidation of a related angucycline intermediate prior to C-ring cleavage and rearrangement in the biosynthesis of thioangucycline by resolving the individual roles of the homologs. Therefore, the novelty justifies publication. Since TacS and TacT are the closest characterized homologs, they should be included in the phylogenetic tree (Fig. 3)

However, the manuscript has a major flaw that should be addressed prior to publication. In the opinion of this referee the key conclusion that LugOV performs C-ring cleavage is not sufficiently supported by the current data. If the authors can't provide further evidence of an active role in C-ring cleavage, they just can conclude that LugOV is essential for further reactions towards C-ring cleaved angucyclines and that LugOV can stabilize the epoxide moiety, which is otherwise prone to reduction or hydrolysis. In this context, the authors should also discuss the role of HrbF, the protein that clades together with LugOV in the phylogeny (Fig. 3), which was suggested to regulate the regiospecificity of oxygenation enzymes, thus having a "chaperoning" rather than an enzymatic role (DOI: 10.1038/ja.2013.96).

In addition, the authors should comment on the initial proposal of LugOV as a "putative dehydrogenase-methyltransferase" by the same group (DOI: 10.1002/anie.201814581, Table S7) likely due to a distant relationship to methyltransferases (a relationship that was also found for the LugOV homolog HrbF; DOI: 10.1038/ja.2013.96). This might be interesting, considering the finding of the authors, that "many of the angucyclinones accumulating in the *lugOV* mutant have an 8-hydroxy substitution rather than the 8-methoxy derivatives, which were predominant in the wild-type".

Apart from that, the manuscript is clear and well written. Minimal problems should be addressed, like the misalignment of the text in Fig. 1 (e.g. LugA–C; 8-O-Methyltetrangomycin etc.) and in the figure legend of Fig. 3.

In line 123 limamycins are referred to as 10–12, with 10 being Pratensilin A. This might be a mistake.

Reviewers' comments:

Reviewer #1 (Remarks to the Author):

Elsayed and co-workers report new genetic and biosynthetic studies on a previously uninvestigated class of angucycline-type oxygenases responsible for cleaving the C ring. Lugdunomycin is an unusual antibiotic that is hypothesized to derive from several unusual oxidative rearrangements. As a result, this manuscript reports new enzymology and begins to unravel the unique oxidative cascade that is responsible for the biosynthesis of the lugdunomycin skeleton. The angucyclines are also highly significant for their plethora of biological activities, which makes this manuscript both very timely and impactful in terms of its results. The manuscript is very well-written and will be of great interest to the community and wider field.

First, the authors carried out phylogenetic analyses of the lugdunomycin oxygenase genes to evaluate their position within the greater context of other oxygenases that are implicated in oxidative ring cleavage. The introduction and phylogenetic analyses are especially helpful for establishing the evolutionary heritage of the pathway.

Secondly, The authors carried out targeted deletion of oxygenases lugOI, lugOII, lugOIII, lugOIV, and lugOV. The authors also carried out genetic complementation studies, either with the lug oxygenase or other angucycline oxygenases, which provided evidence as to the function of the gene product and ruled out polar effects due to the knockout. These studies resulted in strains that collectively produced 12 new angucycline metabolites. Each of these metabolites was rigorously characterized via NMR spectroscopy and high-resolution mass spectrometry. Furthermore, the structural analysis of these metabolites provides important insight into the biosynthetic logic of the lugdunomycin oxidative cascade.

Third, and rather illuminatingly, the authors have employed comprehensive metabolome analysis and molecular networking to deconvolute the complex HPLC-MS/MS data that were generated in the study. This is an essential contribution to the manuscript that aids in the interpretation of the complicated structural data that is presented. Or, stated another way, this is an integral contribution that facilitates the interpretation of the biosynthesis of the plethora of metabolites that are accumulated in these strains.

This manuscript is highly significant - it describes a brand new class of C-ring cleaving oxygenases - which should open the door for the study of similar enzymes discovered in many other angucycline gene clusters. I recommend that this manuscript be accepted to Communications Chemistry following minor corrections.

The comments below are minor corrections that should be addressed before the manuscript is accepted for publication:

>>> Many thanks for the positive remarks on our manuscript.

Page 3, Line 56: The authors introduce the abbreviation "BVO" to refer to Baeyer-Villiger Monooxygenases. Please ensure that this abbreviation is used consistently throughout - Fig. 3 looks like it uses "BVMO" instead of BVO. This should be updated throughout the manuscript.

>>> The abbreviation BVO introduced in line 56 refers to the chemical reaction (Baeyer-Villiger oxidation) rather than the enzymes (Baeyer-Villiger monooxygenases). Baeyer-Villiger monooxygenases or BVMOs were additionally introduced in page 5 line 92, and both were used consistently throughout.

Page 5, Line 35: The others use the spelling "orthologue" here and "ortholog" elsewhere in the manuscript. The spelling should be corrected throughout the manuscript for consistency.

>>> corrected.

Page 7, Line 109: Fig. 3 features five labels under the header "Middle Ring" that are not displaying correctly. 'B-ring cleavage oxygenases', 'Type 'O' BVMO oxygenases', 'Type '1' BVMO oxygenases', 'Type 'II' BVMO oxygenases', and 'Anthrone oxygenases' all appear like the text was corrupted during the conversion of the file to .PDF or .EPS. The authors should re-generate the figure to enhance the legibility of the labels.

>>> Indeed the text was corrupted during file conversion. The figure is re-generated.

Page 8, Lines 136 - 144: The authors make a very significant observation regarding the production of limamycins and lugdunomycin on R5 and MM media. This is very interesting that the production of these different metabolites is media-dependent, and it gives additional weight to the metabolomics data that is presented below!

>>> Thanks! Indeed the analysis of the whole metabolome gives more insight on the data than looking on isolated metabolites.

Page 11, Lines 185 - 191: It is quite gratifying that the knock-out of lugOII resulted in the abrogation of C6-ketoreduction and a significant change in the metabolome, as expected!

>>> We were indeed so happy to see that our data interpretation makes sense.

Page 13, Line 232: Change it's to its

>>> Changed.

Page 17, Lines 309 - 319, Lines 348-353 : The production of compound 23 with a glucoside is interesting. There do not appear to be any glycosyltransferases or other glucosylated lug metabolites (as mentioned by the authors in Lines 348 - 353). It would be helpful for the reader if the authors could provide some sort of mechanistic interpretation for how this glucoside was formed in the discussion section.

>>> Additional discussion on this point was included (pages 18–19, lines 372–377).

Page 17, Lines 310 - 311: Please correct the syntax here regarding the degradation of 23. It is unclear what is meant by

"because upon the acquisition of the 2D NMR spectra minor signals due to another compound has increased in intensity". Is compound 23 sufficiently pure to obtain good quality NMR spectra? The spectra appear sufficiently clean for this compound in the supporting information.

>>> Indeed, the compound is sufficiently pure to obtain good quality NMR spectra. We normally first acquire a ¹H NMR spectrum for the purified compounds to check purity. The ones which show sufficient purity are re-measured for the acquisition of 2D NMR spectra. For some compounds including compound **23**, the first measurement shows a purity of more than 90%, while the second measurement of the same samples shows the purity dropping to around 85% (as shown in Figures S64 and S65). This is because of the unstable nature of these compounds. We rephrased the sentence to make it clearer (lines 327–328).

Page 22, Line 409 - 419: the discussion about Grob-type fragmentation is insightful and would be unique to the lugdunomycin pathway, since this has apparently not been described previously for angucyclines.

>>> Thanks for the positive comment.

Reviewer #2 (Remarks to the Author):

The article by Somayah S. Elsayed, Helga U. van der Heul, Xiansha Xiao, Aleksu Nuutila, Laura R. Baars, Changsheng Wu, Mikko Metsä-Ketelä, Gilles P. van Wezel "Unravelling key enzymatic steps in C-ring cleavage during angucyclines biosynthesis" is significant for angucyclinone biosynthetic pathways clarification and complements recent findings in this field with new experiments. The authors have fully

discussed the history of the subject with all the examples and acknowledged the latest hypotheses on the origin of the rearranged angucyclinones. The supplementary file is well-organized and sufficiently detailed for the readers.

>>> We would like to thank the reviewer for the positive remarks on the manuscript.

I suggest the manuscript can be published in its current form with only one small alteration. The statement that this "study showed for the first time that angucyclinone C-ring cleavage proceeds through epoxidation" is a little bit of a stretch. There are at least two experiment proofs shown in the previous papers (one of them could be cited as well):

1) Fu, X.-Z. et al. / J. Org. Chem., 2022, 87, 23, 15998–16010. (DOI: 10.1021/acs.joc.2c02134)

2) Reference # 26 from the current manuscript (Cao, M. M. et al. / Angew. Chem. Int. Ed., 2021, 60, 7140–7147 (DOI: 10.1002/anie.202015570) states: "Knock-out experiments therein (pages S68-S69) reliably show, that, at least, part of the fragmented angucyclinones (compounds #2,3 in this paper) are formed according to "epoxide hypothesis" and demonstrates the pathway in Figure S30 (https://onlinelibrary.wiley.com/action/downloadSupplement?doi=10.1002/anie.202015570&file=anie202015570-sup-0001-misc_information.pdf).

But this aspect does not diminish the quality of the current manuscript, which is fully suitable for the Communications Chemistry journal. The article will be noticeable in its field providing new evidence in the revision of rearranged angucyclinones biosynthesis.

>>> We agree with the reviewer that our statement may have been worded too strongly. We have rewritten the sentence (line 75) and made additional changes for clarification (lines 438–439 and 480–481).

However, we do still claim that our work is the first to show with experimental evidence that angucyclinone C-ring cleavage proceeds through epoxidation. This is because the paper of Fu et al. does not provide any experimental evidence on the involvement of the epoxidation in angucyclinones C-ring cleavage. On the contrary, based on the propositions put forward by the authors (and many other publications as well), C-ring cleavage might not necessarily require a prior epoxidation step, but alternatively Baeyer–Villiger oxidation, as shown in scheme 1 in the paper.

Cao et al. present some knock-out experiments showing that the production of some C-ring cleaved angucyclinones was abolished upon the deletion of the two genes, *tacS* and *tacT*, which are based on the authors involved in the epoxidation of angucyclinones. We did acknowledge such finding in our manuscript (lines 410–411), and see that it supports our claims. Cao et al.; however, did not investigate the involvement of epoxidation in C-ring cleavage, and accordingly did not make any clear conclusions in their paper on that point. Actually, the chromatograms (Figure S29) and proposed pathway (Figure S30) included in the supplementary information file, which show the knock-out experiments, did not provide information on all the C-ring cleaved angucyclinones identified in the same study.

Reviewer #3 (Remarks to the Author):

The study by Elsayed et al. gives insights into the biosynthesis of lugdunomycin by targeted deletion of five oxygenase genes, in combination with molecular networking and structural elucidation of eleven new metabolites. The key finding is the identification of LugOIII as the epoxidase catalyzing the installation of a C6a–C12a epoxide group, which appears to be the prerequisite for C-ring cleavage. In addition, the authors could show that LugOV is also essential for the biosynthesis of C-ring cleaved angucyclines and can stabilize the epoxide group. Towards this end the data is convincing and the conclusions made by the authors are sound. The findings are an advance to the recent finding that the LugOIII/OV homologs TacS/T are involved in epoxidation of a related angucycline intermediate prior to C-ring cleavage and rearrangement in the biosynthesis of thioangucycline by resolving the individual

roles of the homologs. Therefore, the novelty justifies publication. Since TacS and TacT are the closest characterized homologs, they should be included in the phylogenetic tree (Fig. 3)

>>> We would like to thank the reviewer for the constructive remarks provided. We have reconstructed the phylogenetic tree to include TacS and TacT, which is indeed a good addition.

However, the manuscript has a major flaw that should be addressed prior to publication. In the opinion of this referee the key conclusion that LugOV performs C-ring cleavage is not sufficiently supported by the current data. If the authors can't provide further evidence of an active role in C-ring cleavage, they just can conclude that LugOV is essential for further reactions towards C-ring cleaved angucyclines and that LugOV can stabilize the epoxide moiety, which is otherwise prone to reduction or hydrolysis. In this context, the authors should also discuss the role of HrbF, the protein that clades together with LugOV in the phylogeny (Fig. 3), which was suggested to regulate the regiospecificity of oxygenation enzymes, thus having a "chaperoning" rather than an enzymatic role (DOI: 10.1038/ja.2013.96).

>>> We agree that the statement in the conclusion that LugOV directly performs C-ring cleavage may be an oversimplification. In the discussion section on the enzyme, we phrase it more appropriately, in the sense that LugOV likely directs the biosynthetic pathway towards C-ring cleavage (line 421), and in one of the routes proposed in Figure 7 (Path B), LugOV should not directly result in C-ring cleavage. We have toned down the sentence in the conclusion, and doublechecked the rest of the manuscript.

As for the research previously conducted on the LugOV homolog, HrbF, we believe that it is unlikely for HrbF, and also LugOV, to have such a regulatory role. The authors used relatively insensitive UV spectroscopy only, while we used much more sensitive MS methods, which clearly demonstrated the presence of 5-hydroxy-containing angucyclinones in the wild-type strain harbouring an active LugOV, although at a lower intensity than in the *lugOV* mutant. The reduced intensity of such metabolites in the presence of the active protein resulted in failure to detect them using UV in the case of HrbF. We have added a paragraph to the Discussion (lines 423–433).

In addition, the authors should comment on the initial proposal of LugOV as a "putative dehydrogenase-methyltransferase" by the same group (DOI: 10.1002/anie.201814581, Table S7) likely due to a distant relationship to methyltransferases (a relationship that was also found for the LugOV homolog HrbF; DOI: 10.1038/ja.2013.96). This might be interesting, considering the finding of the authors, that "many of the angucyclinones accumulating in the *lugOV* mutant have an 8-hydroxy substitution rather than the 8-methoxy derivatives, which were predominant in the wild-type".

>>> Indeed one of the putative annotations for LugOV is "dehydrogenase-methyltransferase", but this is a low similarity hit, just like the "antibiotic biosynthesis monooxygenase" annotation. More detailed analysis of LugOV in the current study against the Pfam database identified a possible antibiotic biosynthesis monooxygenase (ABM) domain, although again with low probability. LugOIII has a high similarity to ABM-containing proteins, and alignment of LugOIII and LugOV revealed that the proteins share 30% aa sequence identity (lines 100–103). All in all, we believe that LugOV does not act as a methyltransferase. The increased production of 8-hydroxy angucyclinones in the *lugOV* mutant is likely due to the absence of LugOV that would normally continuously consume the 8-methoxy angucyclinones, its preferred substrates, and thus direct the biosynthetic pathway towards their production. The methylation of the 8-hydroxy group is most likely performed by LugN that is encoded by the *lugdunomycin* BGC.

Apart from that, the manuscript is clear and well written. Minimal problems should be addressed, like the misalignment of the text in Fig. 1 (e.g. LugA–C; 8-O-Methyltetrangomycin etc.) and in the figure legend of Fig. 3.

>>> All figures were doublechecked and adjusted accordingly.

In line 123 limamycins are referred to as 10–12, with 10 being Pratensilin A. This might be a mistake.

>>> Sentence adjusted.

REVIEWERS' COMMENTS:

Reviewer #2 (Remarks to the Author):

The authors provided a well-reasoned response to the comments and acknowledged previous findings in the field with the necessary references. The corrections in the text were made and clarifications for reviewers were provided to accept this manuscript for journal publication.

Reviewer #3 (Remarks to the Author):

The authors have satisfactorily addressed all concerns raised by the reviewers. I fully support publication of the revised manuscript.